# Antimicrobial Activity of Chamomile Essential Oil: Effect of Different Formulations

**DOI:** 10.3390/molecules24234321

**Published:** 2019-11-26

**Authors:** Sourav Das, Barbara Horváth, Silvija Šafranko, Stela Jokić, Aleksandar Széchenyi, Tamás Kőszegi

**Affiliations:** 1Department of Laboratory Medicine, Faculty of Medicine, University of Pécs, H-7624 Pécs, Hungary; pharma.souravdas@gmail.com; 2Institute of Pharmaceutical Technology and Biopharmacy, Faculty of Pharmacy, University of Pécs, H-7624 Pécs, Hungary; barbara.horvath@aok.pte.hu (B.H.); szechenyi.aleksandar@gytk.pte.hu (A.S.); 3Faculty of Food Technology Osijek, University of Osijek, Franje Kuhaca 20, 31000 Osijek, Croatia; silvija.safranko@ptfos.hr (S.Š.); stela.jokic@ptfos.hr (S.J.); 4János Szentágothai Research Center, University of Pécs, H-7624 Pécs, Hungary

**Keywords:** chamomile essential oil, Pickering emulsion, antimicrobial activity, free radical generation

## Abstract

Essential oils (EOs) are highly lipophilic, which makes the measurement of their biological action difficult in an aqueous environment. We formulated a Pickering nanoemulsion of chamomile EO (C_Pe_). Surface-modified Stöber silica nanoparticles (20 nm) were prepared and used as a stabilizing agent of C_Pe._ The antimicrobial activity of C_Pe_ was compared with that of emulsion stabilized with Tween 80 (C_T80_) and ethanolic solution (C_Et_). The antimicrobial effects were assessed by their minimum inhibitory concentration (MIC_90_) and minimum effective (MEC_10_) concentrations. Besides growth inhibition (CFU/mL), the metabolic activity and viability of Gram-positive and Gram-negative bacteria as well as *Candida* species, in addition to the generation of oxygen free radical species (ROS), were studied. We followed the killing activity of C_Pe_ and analyzed the efficiency of the EO delivery for examined formulations by using unilamellar liposomes as a cellular model. C_Pe_ showed significantly higher antibacterial and antifungal activities than C_T80_ and C_Et_. Chamomile EOs generated superoxide anion and peroxide related oxidative stress which might be the major mode of action of Ch essential oil. We could also demonstrate that C_Pe_ was the most effective in donation of the active EO components when compared with C_T80_ and C_Et_. Our data suggest that C_Pe_ formulation is useful in the fight against microbial infections.

## 1. Introduction

Essential oils (EOs) have been widely used in folk medicine throughout the history of humankind. The application of EOs covers a wide range from therapeutic, hygienic, and spiritual to ritualistic purposes. EOs are aromatic, volatile, lipophilic liquids extracted from different parts of plant materials such as barks, buds, flowers, fruits, seeds, and roots [1]. EOs are mixtures of complex compounds with variable individual chemical composition and concentrations that includes primarily terpenoids, like monoterpenes (C10), sesquiterpenes (C15), diterpenes (C20), acids, alcohols, aldehydes, aliphatic hydrocarbons, acyclic esters or lactones, rare nitrogen- and sulfur-containing compounds, coumarin, and homologues of phenylpropanoids [1,2]. The biological effects of EOs cover a wide range of effects, including antioxidant, antimicrobial, antitumor, anti-inflammatory, and antiviral activity [3].

The increase in demand for the use of aromatherapy as complementary and alternative medicine has led people to believe in the myth that EOs are harmless because they are natural and have been used for a long time [4]. However, there might be several side effects of EOs even if topical administration is applied and, among these, allergic reactions are the most frequent (many EOs can cause, e.g., rashes on the skin). Some of them can be poisonous if absorbed through the skin, breathed, or swallowed. Previous studies also report the interaction of EOs with other drugs [5]. The continuous production of new aroma chemicals and their widespread and uncontrolled usage as alternative therapies together with many carrier diluents have brought serious problems, especially among children. In this regard, it is of utmost importance to study the mode of action of essential oils and to find a proper, unharmful formulation. Another serious problem is the highly lipophilic nature of EOs, which makes it impossible to measure their biological effects in an aqueous environment [1,6,7].

One major characteristic and application of EOs are their strong antimicrobial activity, including antibacterial and antifungal effects without the development of microbial resistance. Numerous studies are found in the literature describing the antimicrobial activities of a large variety of EOs [8,9,10,11,12,13]. Most of these assays include conventional broth dilution method, disk diffusion method, and bioautography assay to measure the antimicrobial activity of EOs. Efforts have been made to overcome the lipophilic nature of the oils usually by application of EOs diluted in seemingly suitable solvents/detergents. In the case of natural lipophilic volatile compounds like EOs, solvents of varying polarity, e.g., DMSO, ethanol, and methanol, are most commonly used. However, previous studies have reported the antimicrobial effects of the solvents themselves (DMSO, ethanol, and other solvents in various microbial assays) or their influence on the true antimicrobial effects of EOs [14]. The usage of solubilizing agents limits the precise determination of the antimicrobial activities of EOs. Also, a major problem might arise in the classical assays due to the evaporation of EOs during the assay or the inability of the test microbes to reach the lipophilic range of the tested EOs (in bioautography, as an example) [15,16].

Therefore, new formulations have been determined to increase the solubility or to emulsify the EOs in an aqueous environment. These efforts help to stabilize the oils, to produce an even release of the active components into the required environment, and to maintain their antioxidant and antimicrobial activities [6,11,17,18]. Detergents and organic solvents are not welcome in this regard. Attempts have been made to entrap EOs by modified cyclodextrins for the exact determination of their antimicrobial characteristics [19,20].

The application of Pickering nanoemulsion is a quite novel approach to stabilize oil-in-water (O/W) and water-in-oil (W/O) emulsions by solid particles instead of surfactants. The mechanism involves the adsorption of solid particles on the oil–water interface, causing a significant decrease in the interfacial surface tension that results in high emulsion stability [18]. Previous studies have reported decreased evaporation of EOs from O/W emulsion of nanoparticle-stabilized formulations versus EO–surfactant systems to be a beneficial factor [21,22].

Despite the numerous existing studies on EO–Pickering emulsion, we could not find any literature data on chamomile volatile oil–nanoparticle formulation [7,23]. The main aim of the present work is to use Pickering emulsion of chamomile EO stabilized with modified Stöber silica nanoparticles and characterize its antimicrobial effect using Gram-positive and Gram-negative bacteria as well as *Candida* fungal species. We could demonstrate the strong antimicrobial effects of the chamomile EO–Pickering emulsion and suggest a plausible mode of action of this formulation. Experimental efforts were made to support the suggested mode of action.

## 2. Results

### 2.1. Characteristics of Stöber Silica Nanoparticles

The mean diameter, PDI value (polydispersity index), and zeta potential of modified Stöber silica nanoparticles (SNPs) were determined by dynamic light scattering (DLS), and these values were 20 nm, 0.01, and −21.3 mV, respectively. The size and morphology of SNPs were examined by TEM (see Figure 1). The size distribution obtained by DLS was confirmed by TEM, which showed that the mean diameter of silica samples was 20 nm; they are highly monodisperse and have a spherical shape.

### 2.2. Nanoemulsion Stability

We have prepared a Pickering nanoemulsion with surface-modified silica nanoparticles as a stabilizing agent; the particle concentration was 1 mg/mL in every case. The chamomile EO concentration was 100 µg/mL. To compare properties of chamomile EO–Pickering nanoemulsion (C_Pe_) with the conventional, surfactant-stabilized nanoemulsions, and an emulsion with the Tween 80 stabilizing agent was also prepared. The concentration of surfactant was the same as nanoparticles, 1 mg/mL. The emulsions were stored at room temperature (25 °C).

We considered the emulsion to be stable when its droplet size does not change and sedimentation, aggregation of particles, or phase separation cannot be observed. The results show that the prepared Pickering emulsion is more stable than conventional emulsion (see Table 1). When the volume fraction of chamomile EO was very low, we assumed that all emulsions were of O/W type and this was confirmed by filter paper tests with CoCl_2_ and dye test with Sudan Red G.

### 2.3. Antibacterial and Antifungal Activities (MIC_90_) of Prepared Emulsions

The effect of chamomile Pickering nanoemulsion, conventional emulsion, and essential oil in ethanol on the growth of some foodborne microbes and opportunistic fungi have been evaluated. The C_Pe_ has been shown to have good antibacterial and antifungal activities (MIC_90_) on *Escherichia coli* (*E. coli*) (2.19 µg/mL)*, Pseudomonas aeruginosa* (*P. aeruginosa*) (1.02 µg/mL)*, Bacillus subtilis* (*B. subtilis*) (1.13 µg/mL)*, Staphylococcus aureus* (*S. aureus*) (1.06 µg/mL)*, Streptococcus pyogenes* (*S. pyogenes*) (2.45 µg/mL)*, Schizosaccharomyces pombe* (*S. pombe*) (1.28 µg/mL)*, Candida albicans* (*C. albicans*) (2.65 µg/mL)*,* and *Candida tropicalis* (*C. tropicalis*) (1.69 µg/mL)*,* respectively when compared to C_T80_ counterpart (*P ˂ 0.01*). C_Pe_ showed antimicrobial activity on the selected microbes at an average of fourteen-fold less concentration compared with free essential oil in ethanol (C_Et_). Simultaneously, C_Pe_ showed a similar antifungal effect as caspofungin (Cas) on *Candida tropicalis.* The comparative dose–response curves are shown in Figure 2 and Figure 3 for bacteria and fungi, respectively.

### 2.4. Minimum Effective Concentrations (MEC_10_) for Tested Bacteria and Fungi

The minimum effective concentration (MEC_10_) of C_Pe_, C_T80_, and C_Et_ on foodborne Gram-positive and Gram-negative bacteria as well as fungi have been determined. The dose–response curve shows a slow killing effect (≤10% of the population) of C_Pe_ after 1 h of treatment at a two-fold higher concentration compared with MIC_90_ data. The MEC_10_ also highlights the effective killing effect of C_Pe_ when compared to C_T80_ and C_Et_ (Figure 4 and Figure 5) (*P* ˂ 0.01).

### 2.5. Effect on Microbial Oxidative Balance

Reactive oxygen species (ROS) production and accumulation in the cells initiates oxidative stress, leading to cellular structural damage followed by induced apoptosis [6]. We have investigated the relationship between oxidative stress generation after 1 h of treatment and microbial killing activity. The results are demonstrated in Figure 6 and Figure 7 for bacteria and fungi, respectively. Data expressed as % of the control are as follows: the ROS (1085.86 ± 126.36), peroxide (1229.86 ± 164.52) and superoxide (1276.86 ± 165.42) generation were the highest in case of *S. aureus*. The C_Pe_ showed an effective increment of ROS, peroxide, and superoxide generation in both Gram-positive and -negative bacteria when compared to C_T80_ and C_Et_ (*P* < 0.01). C_Pe_ showed increased oxidative stress in both bacteria and fungi at least seven-fold higher than the negative control whereas the positive control (menadione) produced an eight- to nine-fold increase in 1 h. The C_Et_ has generated a two to four-fold increment in oxidative stress which is the lowest among all tested compounds. 

### 2.6. Time–Kill Kinetics Study

The time–kill kinetics curve was performed to quantify living populations after a definite time interval under different sample MEC_10_ concentrations. A significant reduction (four log-fold) in the cell survivability has been observed in case of C_Pe_ when compared to Gc (*P* < 0.01) (Figure 8 and Figure 9). Fifty percent of cell death occurred by C_Pe_ at 16 and 36 h in the case of bacteria and fungi, and was most effective in reducing living colonies in case of *C. albicans* (1.73 ± 0.15 CFU/mL) after 48 h of treatment. At an average of a two-fold higher concentration, C_Et_ was able to show a killing effect compared to C_Pe_ (*P* < 0.01).

### 2.7. Live/dead Cell Viability Discrimination

The effect of C_Pe_, C_T80_, and C_Et_ on the viability of selected bacteria and fungi were tested (Figure 10 and Figure 11). C_Pe_ decreases the viability of the tested bacteria and fungi with an average viability reduction to 42.36% ± 3.74% and 49.62% ± 5.25% of mean percentage viability compared to Gc after 16 and 36 h of treatments in bacteria and fungi respectively (*P* < 0.01), whereas C_T80_ and C_Et_ were less effective than C_Pe_ with mean percentage viabilities of ≥60% and 70%, respectively.

### 2.8. Interaction Study between Cell Model and Different Formulations of Chamomile EO

The unilamellar liposomes (ULs), consisting of a single phospholipid, can be used as artificial cells or biological membrane model for studying the interactions between cells or cell membranes and drugs or biologically active components [24]. This study was conducted to determine the intracellular delivery ability of active components from chamomile EO for different formulations. 

We have studied the interaction of ULs and different forms of chamomile EO for 24 h at 35 °C. After 1 h of interaction, 27.2% of EO have penetrated the liposomes from Pickering nanoemulsion, while conventional emulsion and the ethanolic solution did not provide a measurable amount. The next sampling was after 2 h, where Pickering emulsion has delivered 48.3% of EO, conventional emulsion 0.5%, while the amount of EO delivered by the ethanolic solution was not measurable. Final sampling was after 24 h, with 82.2% of chamomile EO found in the ULs when it was introduced in Pickering nanoemulsion form, and this value was 66.8% for conventional emulsion and 32.5% for the ethanolic solution (see Table 2).

## 3. Discussion

The effect of three different formulations on antimicrobial activity of chamomile essential oil has been examined. We have successfully prepared stable Pickering nanoemulsion using silica nanoparticles with appropriate lipophilicity. The emulsion was stable for three months. The effectiveness of Pickering emulsion was compared with conventional emulsion and ethanolic solution.

Based on our antimicrobial activity analyses, C_Pe_ shows higher growth inhibitory action and consequently lower MICs compared to C_T80_ and C_Et_. Many researchers have studied the antimicrobial activity of chamomile oil [3,7,13,23,25], however, the mechanism of action at subinhibitory concentrations has not previously been studied. Our data suggest an effective killing activity of C_Pe_ on selected bacteria and fungi. It is believed that EOs act against cell cytoplasmic membrane and induce stress in microorganisms [26,27,28,29]. To visualize the effects of C_Pe_, C_T80_, and C_Et_, we introduced different staining methods to understand their mechanism. C_Pe_ was able to generate higher oxidative stress compared to the conventional emulsion and ethanolic solution followed by metabolic interference and cell wall disruption and finally caused cell death at subinhibitory concentration [21,30,31,32,33].

The results obtained in the model experiment show that C_Pe_ is the most effective form for the intracellular delivery of chamomile EO. Based on these results it can be established that the different antibacterial and antifungal effects may be caused by the difference of adsorption properties of EO forms to the microbial cells. The mechanism of delivery has not been revealed in this study, but evidence for the adsorption of Pickering emulsion droplets on the cell membrane has been previously reported [18]. Assuming the adsorption of C_Pe_ droplets on the cell membrane of investigated microbes, intracellular delivery of active components from EO is feasible in two ways. Passive diffusion is caused by the higher local concentration gradient of EO on the cell membrane, or fusion of C_Pe_ droplets with microbial cells. Overall, our observations demonstrate that C_Pe_ facilitates chamomile oil to permeate cells, inducing oxidative stress and disrupting the membrane integrity because of higher adsorption efficacy of chamomile EO. SNP acts as a stabilizer, inhibiting the easy escape of EOs from the emulsion system compared to the conventional emulsion and free oils.

## 4. Materials and Methods

### 4.1. Synthesis, Surface Modification, and Characterization of Stöber Silica Nanoparticles

We have performed the synthesis of 20 nm hydrophilic silica nanoparticles with the previously reported modified Stöber method [9]. Briefly, a solution of tetraethoxysilane (TEOS) and ultrapure water in ethanol was prepared by using tetraethoxysilane, (Thermo Fisher GmbH, Kandel, Germany, pur. 98%); absolute ethanol AnalaR Normapur ≥99.8% purity (VWR Chemicals, Debrecen, Hungary) and water (membraPure Astacus Analytical with UV, VWR Chemicals, Debrecen, Hungary). The solution was stirred for 20 min and sonicated for another 20 min (Bandelin Sonorex RK 52H, BANDELIN electronic GmbH & Co. KG, Berlin, Germany). An appropriate amount of NH_3_ solution (28% (*w*/*w*) ammonium solution, VWR Chemicals, Debrecen, Hungary) was added to the reaction mixture and was stirred at 1000 rpm for 24 h at room temperature (25 °C). The molar ratio of components was water/ethanol/TEOS/NH_3_ = 100:300:5.2:1. The surface of hydrophilic silica nanoparticles was modified with propyltriethoxysilane (PTES Alfa Aesar, Haverhill, MA, USA, pur. 99%) in a post-synthesis modification reaction [32]. The ethanolic solution of the modifying agent was added to the freshly prepared hydrophilic silica nanoparticle suspensions; the mixtures were stirred for 6 h with 1000 rpm at room temperature. Before further use of the SNPs, the ammonium hydroxide and ethanol were always removed from the reaction mixture by distillation (Heidolph Laborota 4000, Heidolph Instruments GmbH & CO. KG, Germany). The water content was supplemented three times. The concentration of silica nanoparticle water-based suspension was finally adjusted to 1 mg/cm^3^.

The size distribution and zeta potential of silica nanoparticles were determined by dynamic light scattering (DLS) using Malvern Zetasizer NanoS and NanoZ instruments (Malvern Instruments-Malvern Panalytical, Worcester, UK). The morphology and size distribution were also examined with transmission electron microscopy (TEM), (JEOL JEM-1200 EX II and JEM-1400, JEOL Ltd., Tokyo, Japan). The samples were dropped onto 200 mesh copper grids coated with carbon film (EMR Carbon support grids, Micro to Nano Ltd, Haarlem, The Netherlands) from diluted suspensions.

### 4.2. Preparation and Characterization of Pickering Nanoemulsion

As stabilizing agents, surface-modified silica nanoparticles or Tween 80 surfactant (Polysorbate80, Acros Organics, New Jersey, NJ, USA) were used. The concentration of stabilizing agents and chamomile essential oil (bluish *Matricaria chamomilla* oil, Aromax Ltd., Budapest, Hungary) was kept constant for all experiments, the values were 1 mg/mL and 100 µg/mL, respectively. The first step of the emulsification process was sonication for 2 minutes (Bandelin Sonorex RK 52H, BANDELINelectronic GmbH & Co. KG, Germany), then emulsification using UltraTurrax (IKA Werke T-25 basic, IKA®-Werke GmbH & Co. KG, Germany) for 5 min at 21,000 rpm. To compare the different formulations, an ethanolic solution was also prepared; chamomile essential oil was added to absolute ethanol at 100 µg/mL concentration, and the solution was sonicated for 5 min.

The stability of Pickering emulsion was studied from periodical droplet size determination using DLS measurements (Malvern Zetasizer Nano S, Malvern Panalytical Ltd, Worcester, UK).

### 4.3. Materials for Biological Experiments

In these experiments, the sterile 96-well microtiter plates were from Greiner Bio-One (Kremsmunster, Austria), potassium phosphate monobasic, glucose, adenine, 96% ethanol (Et), peptone, yeast extract, agar-agar, and Mueller Hinton agar were from Reanal Labor (Budapest, Hungary), modified RPMI 1640 (contains 3.4% MOPS, 1.8% glucose, and 0.002% adenine), SYBR green I 10,000×, propidium iodide, dihydrorhodamine 123 (DHR 123), 2′,7′-dichlorofluorescin diacetate (DCFDA), dihydroethidine (DHE) and menadione (Me) were from Sigma-Aldrich Chemie GmbH (Steinheim, Germany), disodium phosphate and dimethyl sulfoxide (DMSO) were from Chemolab Ltd. (Budapest, Hungary), sodium chloride from VWR Chemicals (Debrecen, Hungary), potassium chloride was from Scharlau Chemie S.A (Barcelona, Spain), 3-(*N*-morpholino) propanesulfonic acid (MOPS) was from Serva Electrophoresis GmbH (Heidelberg, Germany), caspofungin (Cas) from Merck Sharp & Dohme Ltd (Hertfordshire, UK), vancomycin (Van) from Fresenius Kabi Ltd. (Budapest, Hungary), 0.22 µm vacuum filters from Millipore (Molsheim, France) and the cell spreader was from Sarstedt AG & Co. KG (Numbrecht, Germany). All other chemicals used in the study were of analytical or spectroscopic grade. For fungi, we used an in-house nutrient agar medium [34] while phosphate-buffered saline (PBS, pH 7.4) was from Life Technologies Ltd. (Budapest, Hungary). Highly purified water (<1.0 µS) was applied throughout the studies. 

### 4.4. Determination of Minimum Inhibitory Concentration (MIC_90_)

#### 4.4.1. Microorganisms

*Escherichia coli* (*E. coli*) PMC 201, *Pseudomonas aeruginosa* (*P. aeruginosa*) PMC 103, *Bacillus subtilis* (*B. subtilis*) SZMC 0209, *Staphylococcus aureus* (*S. aureus*) ATCC 29213, *Streptococcus pyogenes* (*S. pyogenes*) SZMC 0119, *Schizosaccharomyces pombe* (*S. pombe*) ATCC 38366, *Candida albicans* (*C. albicans*) ATCC 1001, and *Candida tropicalis* (*C. tropicalis*) SZMC 1368 were obtained from Szeged Microbial Collection, Department of Microbiology, University of Szeged, Hungary (SZMC) and Department of General and Environmental Microbiology, Institute of Biology, University of Pecs, Hungary (PMC).

#### 4.4.2. Antimicrobial Activity Tests

The antibacterial activity of the tested drugs was separately evaluated on *E. coli, P. aeruginosa*, *B. subtilis*, *S. aureus*, and *S. pyogenes* according to our previously published protocol [35]. In brief, bacterial populations of ~10^5^ CFU/mL were inoculated into RPMI media and incubated for 16 h at 35 ± 2 °C with test compounds (C_Pe_, C_T80_, C_Et_, and Van) over a wide concentration range (0.3–0.01 µg/mL). The absorbance was measured by a Thermo Scientific Multiskan EX 355 plate reader (InterLabsystems, Budapest, Hungary) at 600 nm.

The antifungal activity against *S. pombe*, *C. albicans*, and *C. tropicalis* species were also carried out according to our previously published method [35]. Briefly, ~10^3^ cells/mL were incubated for 48 h at 30 ± 2 °C with test compounds (C_Pe_, C_T80_, C_Et_ and Cas) at wide concentration range (20–0.01 µg/mL) in modified RPMI media. The absorbance was measured by a Thermo Scientific Multiskan EX 355 plate reader (InterLabsystems, Budapest, Hungary) at 595 nm. Absorbance values were converted to percentages compared to growth control (~100%) and data were fitted by nonlinear dose–response curve method to calculate the dose producing ≥90% growth inhibition (MIC_90_). All the measurements were performed by applying three technical replicates in six independent experiments. Van and Cas were used as a standard antibacterial and antifungal drug, respectively, throughout the experiments.

### 4.5. Determination of Minimum Effective Concentration (MEC_10_)

The assay has been designed to determine the killing effects of the test compounds for a certain period of time. In brief, a wide concentration range (0.03–80 µg/mL) of the test samples were used to treat ~10^6^ cells/mL for one hour. One milliliter of treated and untreated samples were taken and were diluted 10^5^ times followed by spreading 50 µL samples onto 20 mL nutrient agar media and incubated for 24 h at 35 °C (bacteria) and 30 °C (fungi) for colony-forming unit (CFU/mL) quantification. The data were compared with growth control and positive control (Van for bacteria and Cas for fungi) for percentage mortality (~10% death) determination. It is noteworthy that in the MEC_10_ experiments, the inoculated microbial cell number was 10^3^ times higher than was used in the MIC_90_ determinations (10^6^ vs. 10^3^). However, in both cases, the same formula was used (see Section 4.7). All the measurements were performed by applying three technical replicates in six independent experiments.

### 4.6. Determination of Microbial Oxidative Generation and Killing Activity 

#### 4.6.1. Quantification of Total ROS Generation 

Total ROS generation was assayed according to previously published protocols [27,28,34,36]. Briefly, ~10^6^ cells/mL were collected and centrifuged at 1500 *g* for 5 min and were suspended in PBS. The cells were stained with a 20 mM stock solution of DCFDA in PBS (pH 7.4) to achieve an end concentration of 25 µM, and were incubated at 35 °C (for bacteria) and 30 °C (for fungi) for 30 min in the dark with mild shaking. The cells were centrifuged (Hettich Rotina 420R, Auro-Science Consulting Ltd., Budapest, Hungary) and suspended in RPMI media. The cells were treated with C_Pe_, C_T80_, C_Et_, Van (bacteria), and Cas (fungi) at their respective MEC_10_ concentrations for one hour. The fluorescence signals were recorded at Ex/Em = 485/535 nm wavelengths by a Hitachi F-7000 fluorescence spectrophotometer/plate reader (Auro-Science Consulting Ltd., Budapest, Hungary). The percentage increase in oxidative balance was measured by comparing the signals to those of the growth controls (Gc). Six independent experiments were done with three technical replicates for each treatment.

#### 4.6.2. Detection of Peroxide (O_2_^2−^) and Superoxide Anion (O_2_^•−^) Generation

The previously described protocol was adapted for peroxide [36,37] and superoxide anion radicals [34,38] with modifications. A positive control, Me (0.5 mmol/L as end concentration), C_Pe_, C_T80_, C_Et_, Van (bacteria), and Cas (fungi) at their respective MEC_10_ concentrations were used to treat the cell suspensions (~10^6^ cells/mL) for an hour at 35 °C (bacteria) and 30 °C (fungi) in RPMI media. Thereafter, the cells were centrifuged at 1500 *g* for 5 min at room temperature followed by resuspension of pellets in PBS of the same volume. DHR 123 (10 µmol/L, end concentration) and DHE (15 µmol/L, end concentration) were added separately to the cell samples for peroxide and superoxide determination. The stained cells were further incubated at 35 °C (bacteria) and 30 °C (fungi) in the dark with mild shaking. The samples were centrifuged and resuspended in PBS followed by the distribution of the samples into the wells of 96-well microplates. The fluorescence was measured at excitation/emission wavelengths of 500/536 nm for peroxides and 473/521 nm for superoxide detection by a Hitachi F-7000 fluorescence spectrophotometer/plate reader (Auro-Science Consulting Ltd., Budapest, Hungary). The percentage increase in oxidative stress was measured by comparing the signals to those of the growth controls (Gc). Six independent experiments were done with three technical replicates for each treatment.

#### 4.6.3. Time–Kill Kinetics Assay

We followed a protocol previously published by T. Appiah et. al., with modifications [39]. In brief, C_Pe_, C_T80_, C_Et_, Van (bacteria), and Cas (fungi) at their respective MEC_10_ concentrations were used to treat the microbial population of ~10^6^ CFU/mL and were incubated at 35 °C (bacteria) and 30 °C (fungi). One milliliter of the treated and untreated samples was pipetted at time intervals of 0, 2, 6, 8, 16, and 24 h for bacteria, and 0, 6, 12, 30, 36, and 48 h for fungi, and were diluted 10^5^ times followed by spreading 50 µL onto 20 mL nutrient agar media using a cell spreader and incubated at 35 °C (bacteria) and 30 °C (fungi) for 24 h. Van and Cas were used as reference controls for bacteria and fungi. Control without treatment was considered as growth control (Gc). The colony-forming unit (CFU/mL) of the microorganisms were determined, performed in triplicate and was plotted against time (h). Six independent experiments were done with three technical replicates for each treatment.

#### 4.6.4. Live/dead Discrimination of Microbial Cells

For live/dead cell discrimination, we followed the protocol published previously [35]. In brief, the cell population of ~10^6^ cells/mL were treated with C_Pe_, C_T80_, C_Et_, Van (bacteria), and Cas (fungi) at their respective MEC_10_ concentrations and were incubated at 35 °C (bacteria) and 30 °C (fungi). Treated and untreated samples were pipetted at time intervals of 0, 2, 6, 8, 16, and 24 h for bacteria, and 0, 6, 12, 30, 36 and 48 h for fungi followed by centrifugation at 1000 *g* for 5 min, washed, and resuspended in PBS (100 µL/well). One hundred microliters of freshly prepared working dye solution in PBS (using 20 µL SYBR green I and 4 µL propidium iodide diluted solutions as described earlier) were added to the samples. The plate was incubated at room temperature for 15 min in the dark with mild shaking. A Hitachi F-7000 fluorescence spectrophotometer/plate reader (Auro-Science, Consulting Ltd., Budapest, Hungary) was used to measure the fluorescence intensities of SYBR green I (excitation/emission wavelengths: 490/525 nm) and propidium iodide (excitation/emission wavelengths: 530/620 nm), respectively. A green to red fluorescence ratio for each sample and for each dose was achieved and the % of dead cells with the response to the applied dose was plotted against the applied test compound doses using a previously published formula [35]. All treatments were done in triplicates and six independent experiments were performed.

### 4.7. Statistical Analysis of Microbiological Experiments

All data were given as mean ± SD. Graphs and statistical analyses were conducted using OriginPro 2016 (OriginLab Corp., Northampton, MA, USA). All experiments were performed independently six times and data were analyzed by one-way ANOVA test. *P* ˂ 0.01 was considered statistically significant. The growth inhibition concentration (MIC_90_) and minimum effective concentration (MEC_10_) were calculated using a nonlinear dose–response sigmoidal curve function as follows:
$$y=A_{1}+\frac{A_{2}-A_{1}}{1+10^{\left(\log_{x}0-x\right)p}}$$ where A1, A2, LOG_x_0 and p as the bottom asymptote, top asymptote, center, and hill slope of the curve have been considered.

### 4.8. Interaction Study between the Cell Model (Unilamellar Liposomes) and Different Formulations of Chamomile EO

Unilamellar liposomes (ULs) have been prepared from phosphatidylcholine (Phospolipon 90G, Phospholipid GmbH, Berlin, Germany) by the modified method described before by Alexander Moscho et al. [40]. Phosphatidylcholine was dissolved in chloroform (≥98% stabilized, VWR Chemicals, Debrecen, Hungary) in 0.1 M concentration, and 150 µL of this solution was diluted in a mixture of 6 mL chloroform and 1 mL of methanol. This solution was added dropwise to 40 mL of PBS buffer while stirring on a magnetic stirrer at 600 rpm (VELP Scientifica Microstirrer, Magnetic Stirrer, Usmate Velate MB, Italy). The solvents were removed on a rotational evaporator at 40 °C (Heidolph Laborota 4000, Heidolph Instruments GmbH & CO. KG, Germany). The resulting suspension volume was set to 25 mL with PBS buffer and stored in the refrigerator at 8 °C until further use. A 5 mL suspension of ULs was mixed with 3 mL Pickering nanoemulsion, conventional emulsion, or ethanolic solution, and the chamomile EO concentration was 100 µg/mL for the different formulations. The mixture was stirred at 600 rpm for 24 h at 35 °C, and 1 mL aliquots were taken after 1, 2, and 24 h. The samples were centrifuged at 3000 rpm and 20 °C for 5 min, and the ULs were collected and dissolved in absolute ethanol. The chamomile–EO content of samples was determined with UV/VIS Spectroscopy at 290 nm (Jasco V-550 UV/VIS Spectrophotometer; Jasco Inc., Easton, MD, USA). For UV/VIS measurements we have prepared samples without chamomile–EOs, i.e., ULs with SNP suspension, Tween 80 solution, or ethanol were also mixed and centrifuged and were used as blanks.

### 4.9. GC-MS Analysis of Chamomile EO

Gas chromatography and mass spectrometry (GC–MS) analyses were carried out on an Agilent Technologies (Palo Alto, CA, USA) gas chromatograph model 7890A with 5975C mass detector. Operating conditions were as follows: column HP-5MS (5% phenylmethyl polysiloxane), 30 m × 0.25 mm i.d., 0.25 µm coating thickness. Helium was used as the carrier gas at 1 mL/min, injector temperature was 250 °C. HP-5MS column temperature was programmed at 70 °C isothermal for 2 min, and then increased to 200 °C at a rate of 3 °C/min and held isothermal for 18 min. The split ratio was 1:50, ionization voltage 70 eV; ion source temperature 230 °C; mass scan range: 45–450 mass units. The percentage composition was calculated from the GC peak areas using the normalization method (without correction factors). The component percentages were calculated as mean values from duplicate GC–MS analyses of the oil sample. The results of GC–MS analysis can be seen in Appendix A.

## Figures and Tables

**Figure 1 molecules-24-04321-f001:**
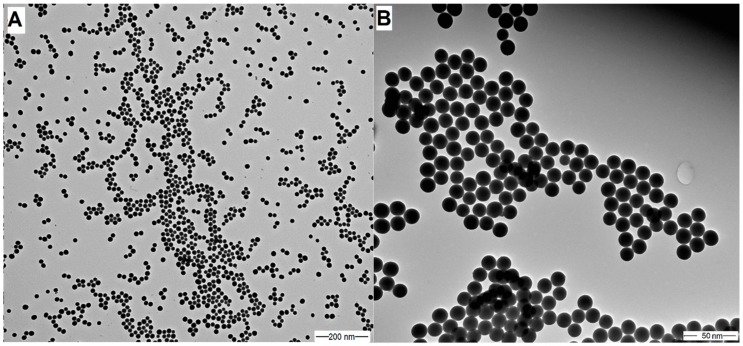
TEM images of silica nanoparticles (SNPs) with different resolutions: 100,000× magnification (**A**) and 500,000× magnification (**B**), accelerating voltage: 80 kV; d_TEM_ = 20 nm. PDI = 0.015.

**Figure 2 molecules-24-04321-f002:**
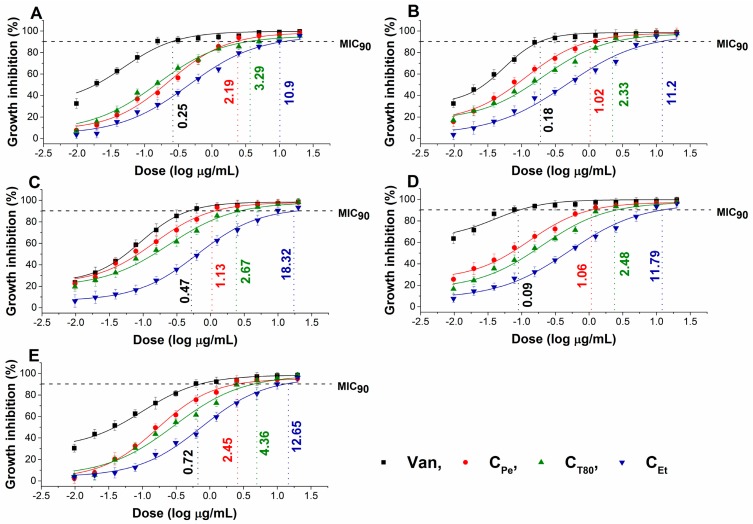
Minimum inhibitory concentration (MIC_90_) of C_Pe_, C_T80_, C_Et_, and vancomycin (Van, µg/mL) on *E. coli* (**A**), *S. aureus* (**B**), *B. subtilis* (**C**), *P. aeruginosa* (**D**), and *S. pyogenes* (**E**).

**Figure 3 molecules-24-04321-f003:**
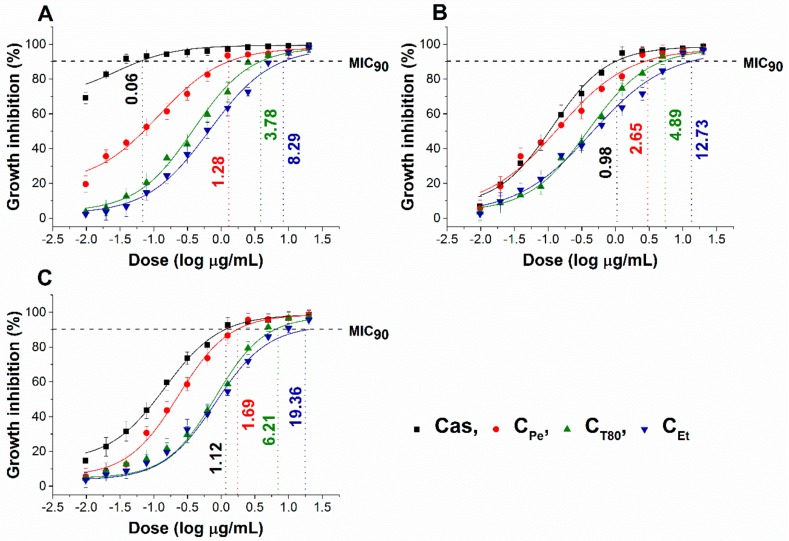
Minimum inhibitory concentration (MIC_90_) of C_Pe_, CT_80_, C_Et_, and caspofungin (Cas, µg/mL) on *S. pombe* (**A**), *C. albicans* (**B**), and *C. tropicalis* (**C**).

**Figure 4 molecules-24-04321-f004:**
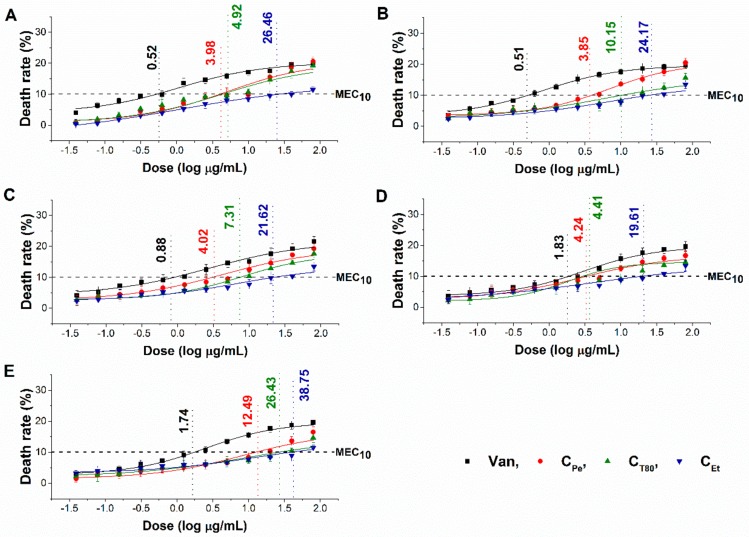
Minimum effective concentration (MEC_10_) of C_Pe_, C_T80_, C_Et_, and Van (µg/mL) on *E. coli* (**A**), *S. aureus* (**B**), *B. subtilis* (**C**), *P. aeruginosa* (**D**), and *S. pyogenes* (**E**).

**Figure 5 molecules-24-04321-f005:**
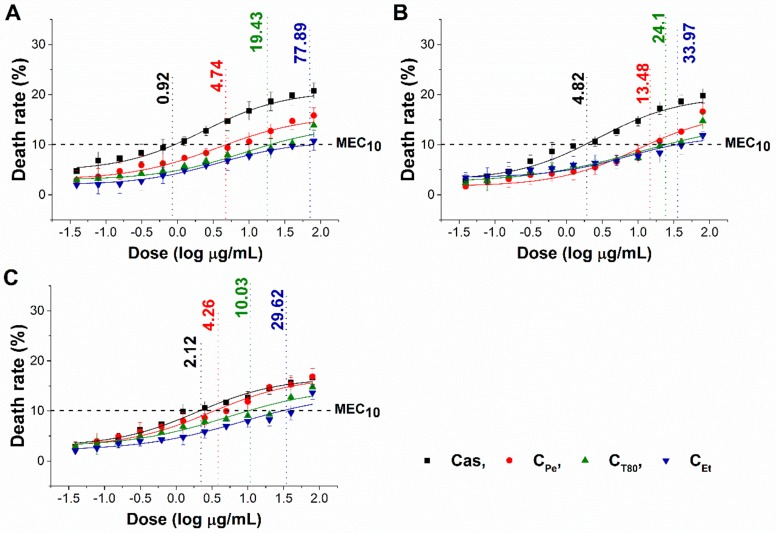
Minimum effective concentration (MEC_10_) of C_Pe_, C_T80_, C_Et_, and Cas (µg/mL) on *S. pombe* (**A**), *C. albicans* (**B**), and *C. tropicalis* (**C**).

**Figure 6 molecules-24-04321-f006:**
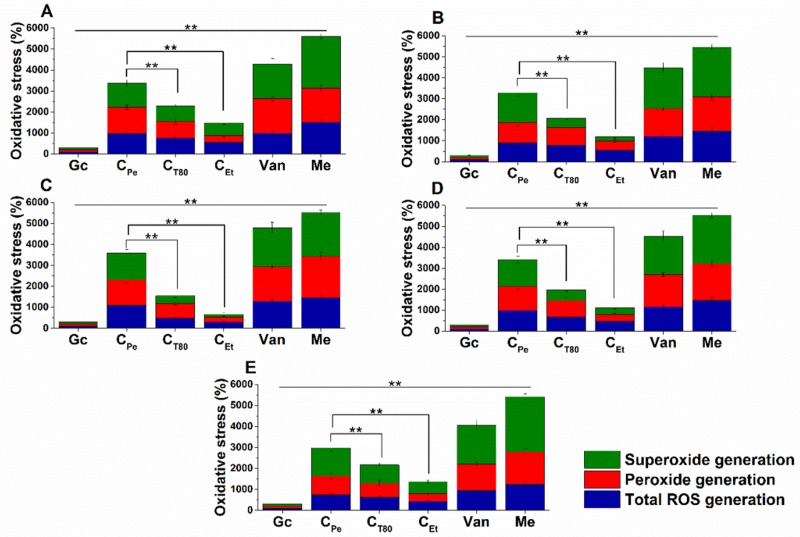
Percentage oxidative stress generation by C_Pe_, C_T80_, C_Et_, and Van on *E. coli* (**A**), *S. aureus* (**B**), *B. subtilis* (**C**), *P. aeruginosa* (**D**), and *S. pyogenes* (**E**). Six independent experiments, each with 3 replicates, compared with menadione (Me) and growth control (Gc) as controls after 1 h of treatment (***P* < 0.01).

**Figure 7 molecules-24-04321-f007:**
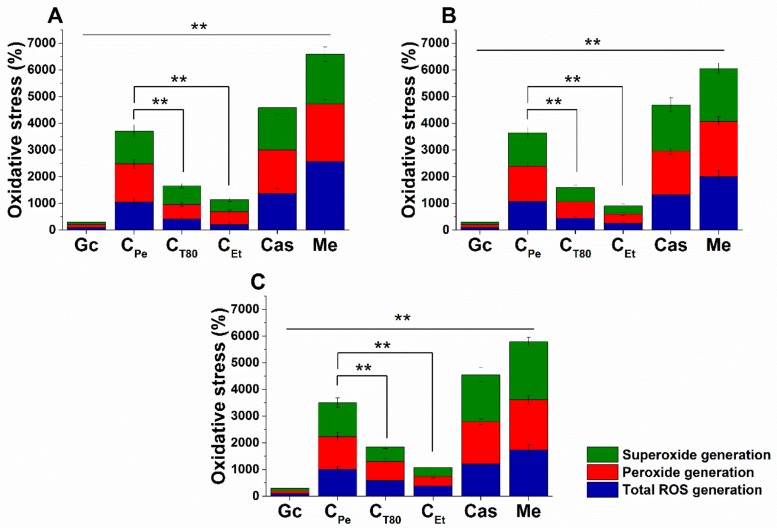
Percentage oxidative stress generation by C_Pe_, C_T80_, C_Et_, and Cas on *S. pombe* (**A**), *C. albicans* (**B**), and *C. tropicalis* (**C**). Six independent experiments, each with 3 replicates, compared with Me and Gc as positive and growth controls after 1 h of treatment (***P* < 0.01).

**Figure 8 molecules-24-04321-f008:**
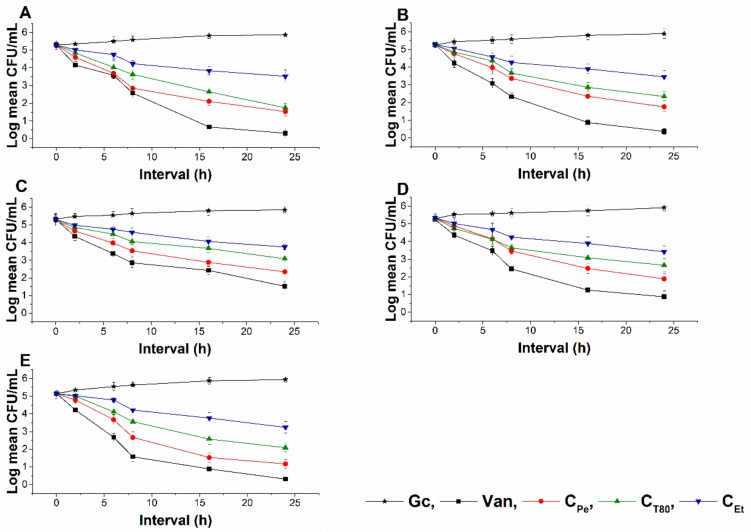
Colony-forming unit (CFU/mL) of C_Pe_, C_T80_, and C_Et_ on *E. coli* (**A**), *S. aureus* (**B**), *B. subtilis* (**C**), *P. aeruginosa* (**D**), and *S. pyogenes* (**E**). Six independent experiments, each with 3 replicates, compared with Van and Gc as positive and growth controls.

**Figure 9 molecules-24-04321-f009:**
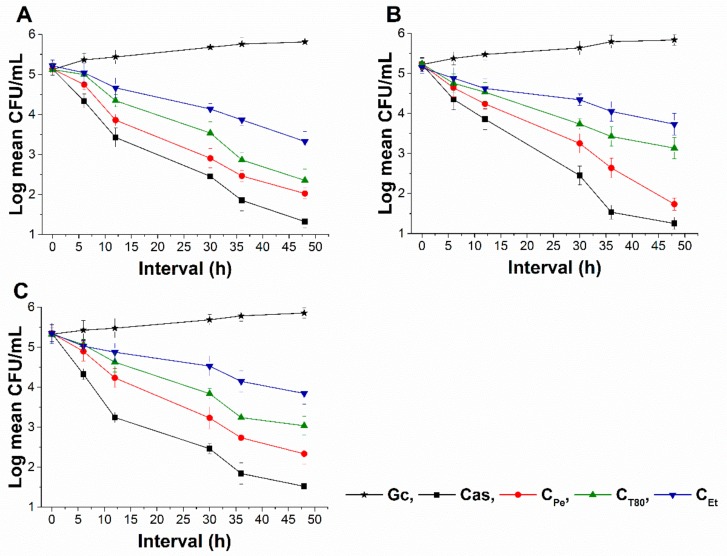
Colony-forming unit (CFU/mL) of C_Pe_, C_T80_, and C_Et_ on *S. pombe* (**A**), *C. albicans* (**B**), and *C. tropicalis* (**C**). Six independent experiments, each with 3 replicates, compared with Cas and Gc as positive and growth controls.

**Figure 10 molecules-24-04321-f010:**
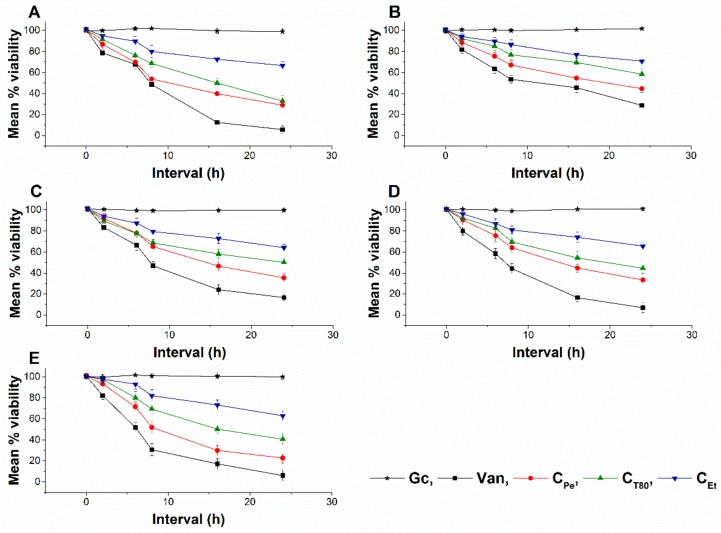
Mean percentage viability of C_Pe_, C_T80_, and C_Et_ on *E. coli* (**A**), *S. aureus* (**B**), *B. subtilis* (**C**), *P. aeruginosa* (**D**), and *S. pyogenes* (**E**). Six independent experiments, each with 3 replicates, compared with Van and Gc as positive and growth controls.

**Figure 11 molecules-24-04321-f011:**
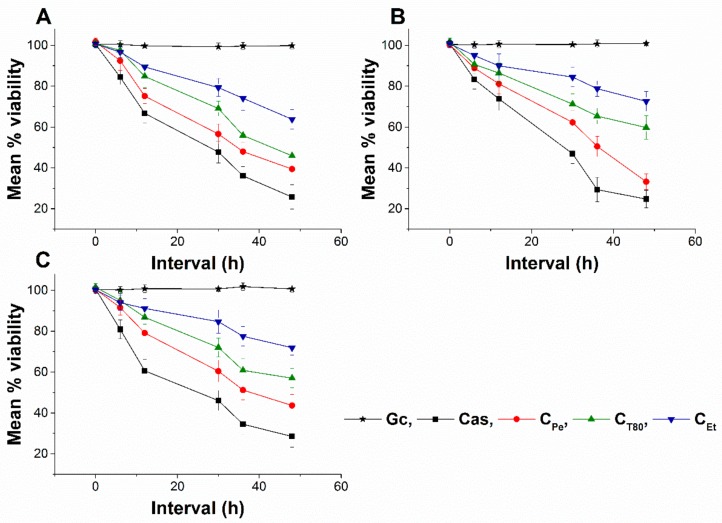
Mean percentage viability of C_Pe_, C_T80_, and C_Et_ on *S. pombe* (**A**), *C. albicans* (**B**) and *C. tropicalis* (**C**). Six independent experiments, each with 3 replicates, compared with Cas and Gc as positive and growth controls.

**Table 1 molecules-24-04321-t001:** Parameters of Pickering nanoemulsion and conventional emulsion.

Stabilizing Agent	D_droplet_ (nm)	Stability
SNP	290 ± 4.5	3 months
Tw80	210 ± 10.5	1 month

**Table 2 molecules-24-04321-t002:** Results of the interaction study between unilamellar liposomes (Uls) and different formulations of chamomile EO. PA means the amount in the percentage of penetrated chamomile EO. Standard deviations were calculated from 3 independent experiments.

Formulation	PA_1h_	PA_2h_	PA_24h_
C_Pe_	27.2 ± 3.7	48.3 ± 5.1	82.2 ± 4.9
C_T80_	-	0.5 ± 0.1	66.8 ± 3.6
C_Et_	-	-	35.5 ± 1.0

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
