# Peer review of "Antimicrobial Activity of Chamomile Essential Oil: Effect of Different Formulations"

_molecules, 2019, doi:10.3390/molecules24234321_

Round 1

Reviewer 1 Report

The manuscript is well written however minor revisions are required

Lines 288-291

It is not clear how long the bacteria were in contact with the treatments, or if they were added from the beginning and then incubated. Please clarify.

Lines 289-293

Why a different concentration was used for bacteria (105) than for yeasts (103)?

Lines 128-130,  Lines 302-307

It is not clear what it refers to "Killing effect on 10% of the population, since a concentration greater than MIC90  was used. The techniques used are different but If the reduction of 1 LOG-UFC corresponds to 90% of the population, how much is a 10% of reduction? How was it calculated with respect to control and how much is it in the microbial population? Clarify,and include the formula used for the calculation.

Lines 302-307, Lines 340-342

No serial dilutions were made? it was only plated directly on the plate? It is possible for count?

Line 212

Cpe shows Higher MICs? or Lower MICs?

Author Response

Response

Reviewer 1

We thank for our reviewer for the devoted work he/she has done and for the useful suggestions. The answers to the questions/comments are below.

The manuscript is well written however minor revisions are required

Lines 288-291

It is not clear how long the bacteria were in contact with the treatments, or if they were added from the beginning and then incubated. Please clarify.

Answer: In brief, bacterial populations of ~105 CFU/mL were inoculated into RPMI media and incubated for 16 h at 35 ± 2 °C with test drugs (CPe, CT80, CEt and Van) at wide concentration range (0.3-0.01 µg/mL). The absorbance was measured by a Multiskan EX 355 (Thermo Electron Corporation, Hungary) at 600 nm.

The antifungal activity against S. pombe, C. albicans and C. tropicalis species were also carried out according to our previously published method [34]. Briefly, ~103 cells/mL were incubated for 48 h at 30 ± 2 °C with test drugs (CPe, CT80, CEt and Cas) at wide concentration range (20-0.01 µg/mL) in modified RPMI media. The absorbance was measured by a Multiskan EX 355 (Thermo Electron Corporation, Hungary) spectrophotometer at 595 nm.

Lines 289-293

Why a different concentration was used for bacteria (105) than for yeasts (103)?

Answer: Regarding the applied concentrations for fungi we would like to refer to the recommendations of NCCLS (M27-A2, vol. 22, No 15.). For bacteria the recommended microbial concentration is higher than for fungi (CLSI M07-A9, vol. 32, No. 2.).

Lines 128-130,  Lines 302-307

It is not clear what it refers to "Killing effect on 10% of the population, since a concentration greater than MIC90  was used. The techniques used are different but If the reduction of 1 LOG-UFC corresponds to 90% of the population, how much is a 10% of reduction? How was it calculated with respect to control and how much is it in the microbial population? Clarify, and include the formula used for the calculation.

Answer: The question is answered in section 4.5. One mL of treated and untreated samples were taken and were diluted 105 times followed by spreading 50 µL samples onto 20 mL nutrient agar media and incubated for 24 h at 35 â—‹C (bacteria) and 30 â—‹C (fungi) for colony-forming unit (CFU/mL) quantification. The data were compared with growth control and positive control (Van, for bacteria and Cas for fungi) for percentage mortality (~10% death) determination. It is noteworthy, that in the MEC10 experiments the inoculated microbial cell number was 103 times higher than was used in the MIC90 determinations (106 vs. 103). However, in both cases the same formula was used (see section 4.7.).

Lines 302-307, Lines 340-342

No serial dilutions were made? it was only plated directly on the plate? It is possible for count?

Answer: Please, see the answer above for lines 302-307. Definitely, the samples were diluted as described below.

One mL of the treated and untreated samples was pipetted at time intervals of 0, 2, 6, 8, 16, and 24 h for bacteria, and 0, 6, 12, 30, 36 and 48 h for fungi and were diluted 105 times followed by spreading 50 µL onto 20 mL nutrient agar media using a cell spreader and incubated at 35 â—‹C (bacteria) and 30 â—‹C (fungi) for 24 h. Van and Cas were used as reference controls for bacteria and fungi. Control without treatment was considered as growth control (Gc). Colony-forming unit (CFU/mL) of the microorganisms were determined, performed in triplicate and was plotted against time (h).

Line 212

Cpe shows Higher MICs? or Lower MICs?

Answer: For better clarity the sentence was rephrased as found below.

Based on our antimicrobial activity analyses, CPe shows higher growth inhibitory action, consequently lower MICs compared to CT80 and CEt.

Reviewer 2 Report

Please consider the following suggestions and comments.

Page 1, Line 17. It appears that an article is missing before the word aqueous. Consider adding an article (an).

Page 1, Line 17. Consider replacing "We formulated Pickering nanoemulsions" with "We formulated a Pickering nanoemulsion".

Page 1, Line 18. It appears that surface modified is missing a hyphen. Consider adding the hyphen (-).

Page 1, Line 18. It seems that agent may not agree in number with other words in this phrase. Consider adding an article (a) before the word stabilizing.

Page 1, Line 20. Consider adding the following words minimum inhibitory before MIC90 and put MIC90 inside parentheses.

Page 1, Line 22. It appears that Gram positive is missing a hyphen. Consider adding the hyphen (-).

Page 1, Line 22. It appears that Gram negative is missing a hyphen. Consider adding the hyphen (-).

Page 1, Line 23. It appears that the singular verb was does not agree with the plural compound subject metabolic activity and viability of Gram positive, Gram negative bacteria and Candida species, the generation of oxygen free radical species (ROS). Consider changing the verb to the plural form (were).

Page 1, Line 25. It appears that an article is missing before the word cellular. Consider adding an article (a).

Page 1, Line 28. The noun phrase donation seems to be missing a determiner before it. Consider adding an article (a).

Page 1, Line 36. Consider changing the noun the mixture to the plural form (mixtures).

Page 1, Line 43. It seems that preposition use may be incorrect here. Consider changing the preposition of with for.

Page 2, Line 53. It seems that there is a pronoun problem here. Consider adding the word it before impossible.

Page 2, Line 56. It appears that an article is missing before the word development. Consider adding an article (the).

Page 2, Line 59. Consider deleting the word techniques.

Page 2, Line 61. It seems that determiner use may be incorrect here. Consider adding the word the before case.

Page 2, Line 65. The spelling of solubilising is a non-American variant. For consistency, consider replacing it with the American English spelling (solubilizing).

Page 2, Line 73. It seems that determiner use may be incorrect here. Consider adding the word the before exact.

Page 2, Line 75. The spelling of stabilise is a non-American variant. For consistency, consider replacing it with the American English spelling (stabilize).

Page 2, Line 76. It appears that water in oil is missing two hyphens. Consider adding these hyphens (-).

Page 2, Lines 78-80. Consider rewriting this sentence for a better understanding of its meaning (Previous studies have reported as a beneficial effect the decreased evaporation from o/w emulsions for nanoparticle-stabilized emulsions versus EOs-surfactant systems [21,22]).

Page 2, Line 86. Consider deleting s from nanoemulsions and replacing one of the plausible modes of actions with a plausible mode of action.

Page 2, Line 89. Consider changing the title of subchapter 2.1 since is similar with the title with subchapter 4.1.

Page 2, Line 92. Consider adding (see 1. Figure) after TEM.

Page 3, Line 96. It seems that resolution may not agree in number with other words in this phrase. Consider changing the noun to the plural form (resolutions).

Page 3, Line 97. Consider changing the title of subchapter 2.2 since is similar with the title with subchapter 4.2.

Page 3, Line 98. Consider adding a before Pickering nanoemulsion.

Page 3, Line 100. The phrase In order to may be wordy. Consider changing In order to with To.

Page 3, Line 101. Consider deleting s from ones, adding an before emulsions, deleting s from emulsions, and adding the before Tween80.

Page 3, Line 110. Consider deleting s from nanoemulsions.

Page 3, Line 111. Consider adding to the title of prepared emulsions after (MIC90).

Page 3, Line 113. The noun phrase growth seems to be missing a determiner before it. Consider adding an article (the).

Page 3, Line 119. Consider adding a before the word similar.

Page 5, Line 127. Consider changing the title of subchapter 2.4 since is similar with the title with subchapter 4.5.

Page 5, Line 128. Consider deleting the space between food and born (foodborne).

Page 5, Line 129. It appears that the singular verb has does not agree with the plural compound subject bacteria and fungi. Consider changing the verb to the plural form (have).

Page 5, Line 129. It appears that dose response is missing a hyphen. Consider adding the hyphen (-).

Page 5, Line 130. It seems that determiner use may be incorrect here. Consider adding a before two-fold.

Page 5, Line 132. It seems that figure may not agree in number with other words in this phrase. Consider changing the noun to the plural form (figures).

Page 6, Line 143. It seems that figure may not agree in number with other words in this phrase. Consider changing the noun to the plural form (figures).

Page 6, Line 145. It appears that the singular verb was does not agree with the plural compound subject the ROS (1085.86 ± 126.36), peroxide (1229.86 ± 164.52) and superoxide (1276.86 ± 165.42) generation. Consider changing the verb to the plural form (were).

Page 6, Line 145. The noun phrase effective increment seems to be missing a determiner before it. Consider adding an article (an).

Page 6, Line 149. It seems that determiner use may be incorrect here. Consider adding the word an before eight to nine-fold.

Page 6, Line 149. It seems that determiner use may be incorrect here. Consider adding the word a before two to four-fold.

Page 7, Line 154. Consider replacing 60 minutes with 1h.

Page 8, Line 161. It appears that time kill is missing a hyphen. Consider adding the hyphen (-).

Page 8, Line 163. It appears that an article is missing before the word case. Consider adding an article (the).

Page 8, Line 163. It seems that figure may not agree in number with other words in this phrase. Consider changing the noun to the plural form (figures).

Page 8, Line 165. It appears that an article is missing before the word case. Consider adding an article (the).

Page 8, Line 166. It seems that determiner use may be incorrect here. Consider adding a before two-fold.

Page 8, Line 166. It appears that an article is missing before the word killing. Consider adding an article (the).

Page 8, Line 169. It appears that colony forming is missing a hyphen. Consider adding the hyphen (-).

Page 9, Line 173. It appears that colony forming is missing a hyphen. Consider adding the hyphen (-).

Page 9, Line 177. It seems that figure may not agree in number with other words in this phrase. Consider changing the noun to the plural form (figures).

Page 11, Line 196. It appears that you are missing a comma after the introductory phrase After 1 hour of interaction. Consider adding a comma.

Page 11, Line 197. The noun phrase ethanolic solution seems to be missing a determiner before it. Consider adding an article (the).

Page 11, Line 199. The noun phrase amount seems to be missing a determiner before it. Consider adding an article (the).

Page 11, Line 202. The noun phrase ethanolic solution seems to be missing a determiner before it. Consider adding an article (the).

Page 11, Line 202. It appears that you are missing a point at the end of the sentence. Consider adding a point.

Page 11, Line 204. It appears that an article is missing before the word amount. Consider adding an article (the).

Page 11, Line 204. The noun phrase percentage seems to be missing a determiner before it. Consider adding an article (the).

Page 11, Line 207. It seems that determiner use may be incorrect here. Consider adding the word the before Effect.

Page 11, Line 214. It seems that the verb have does not agree with the subject. Consider changing the verb to the singular form (has).

Page 11, Line 215. It appears that an article is missing before the word effective. Consider adding an article (an).

Page 11, Line 218. The noun phrase conventional emulsion seems to be missing a determiner before it. Consider adding an article (the).

Page 11, Line 222. It seems that determiner use may be incorrect here. Consider adding the word the before intracellular.

Page 11, Line 228. The noun phrase higher local concentration gradient seems to be missing a determiner before it. Consider adding an article (the).

Page 11, Line 228. The noun phrase cell membrane seems to be missing a determiner before it. Consider adding an article (the).

Page 11, Line 231. The noun phrase easy escape seems to be missing a determiner before it. Consider adding an article (the).

Page 11, Line 232. The noun phrase conventional emulsion seems to be missing a determiner before it. Consider adding an article (the).

Page 12, Line 235. It appears that an article is missing before the word modified. Consider adding an article (the).

Page 12, Line 239. Consider replacing the word additional with another.

Page 12, Line 241. Consider replacing 28w/w% with 28% (w/w).

Page 12, Line 243. Please verify that this propiltriethoxysilane is the correct name of the reagent used.

Page 12, Line 245. The noun phrase Ethanolic solution seems to be missing a determiner before it. Consider adding an article (the).

Page 12, Line 245. The noun phrase modifying agent seems to be missing a determiner before it. Consider adding an article (the).

Page 12, Line 247. It appears that the singular verb was does not agree with the plural compound subject the ammonium hydroxide and ethanol. Consider changing the verb to the plural form (were).

Page 12, Line 247. The noun phrase reaction mixture seems to be missing a determiner before it. Consider adding an article (the).

Page 12, Line 248. Consider adding the acquisition country of the equipment used and a closing bracket after it.

Page 12, Line 256. Consider changing the title of subchapter 4.2 so as to include the preparation of conventional emulsion and ethanolic solution.

Page 12, Line 257. It appears that surface modified is missing a hyphen. Consider adding the hyphen (-).

Page 12, Line 261. It appears that an article is missing before the word emulsification. Consider adding an article (the).

Page 12, Line 272. Vancomycin (Van) was not mentioned in this section (4.3).

Page 12. Nutrient agar media was not mentioned in this section (4.3).

Page 12. PBS was not mentioned in this section (4.3).

Page 12. You did not describe how you prepared CEt (mentioned in the abstract).

Page 13, Line 278. Consider adding 90 after the word MIC.

Page 13, Line 287. The plural verb were does not appear to agree with the singular subject activity. Consider changing the verb to the singular form (was).

Page 13, Line 290. Consider adding the acquisition country of the equipment used.

Page 13, Line 291. Consider mentioning the test drugs used and their concentration range.

Page 13, Line 292. The plural verb were does not appear to agree with the singular subject activity. Consider changing the verb to the singular form (was).

Page 13, Line 294. It seems that ranges may not agree in number with other words in this phrase. Consider changing the noun to the singular form (range).

Page 13, Line 294. Consider mentioning the test drugs used and their concentration range.

Page 13, Line 303. Consider mentioning the test samples used. It seems that the verb were does not agree with the subject. Consider changing the verb to the singular form (was).

Page 13, Line 305. It appears that colony forming is missing a hyphen. Consider adding the hyphen (-).

Page 13. The number of replicates for each sample was not mentioned in this section (4.5).

Page 13, Line 314. Consider adding the model, the producer and the acquisition country of the equipment used (for centrifuge).

Page 13, Line 317. Consider adding the acquisition country of the equipment used.

Page 14, Line 321. It seems that determiner use may be incorrect here. Consider adding the word the before previously.

Page 14, Line 326. It appears that the phrase same volume does not contain the correct article usage. Consider adding the word the before same volume.

Page 14, Line 329. It seems that determiner use may be incorrect here. Consider adding the word the before distribution.

Page 14, Line 342. The plural verb were does not appear to agree with the singular subject Control. Consider changing the verb to the singular form (was).

Page 14, Line 343. It appears that colony forming is missing a hyphen. Consider adding the hyphen (-).

Page 14, Line 343. It seems that the verb were does not agree with the subject. Consider changing the verb to the singular form (was).

Page 14, Line 347. The noun phrase cell population seems to be missing a determiner before it. Consider adding an article (the).

Page 14, Line 352. Consider adding how many hundred microliters were used. It is one hundred?

Page 14, Line 353. It seems that the verb was does not agree with the subject. Consider changing the verb to the plural form (were).

Page 14, Line 355. Consider adding the acquisition country of the equipment used.

Page 14, Line 365. Consider replacing by one-way ANOVA test with using the One-Way ANOVA test.

Page 14, Line 365. It appears that P value is missing a hyphen. Consider adding the hyphen (-).

Page 15, Line 374. Consider adding the acquisition country.

Page 15, Line 376. The noun phrase mixture seems to be missing a determiner before it. Consider adding an article (a).

Page 15, Line 378. The noun phrase magnetic stirrer seems to be missing a determiner before it. Consider adding an article (a).

Page 15, Line 378. Consider adding the model, the producer and the acquisition country of the equipment used (for magnetic stirrer).

Page 15, Line 378. It seems that determiner use may be incorrect here. Consider adding the word a before rotational.

Page 15, Line 378. Consider adding the model, the producer and the acquisition country of the equipment used (for rotational evaporator).

Page 5, Line 379. It seems that preposition use may be incorrect here. Consider changing the preposition on with at.

Page 15, Line 393. Consider moving 0.25 µm before coating thickness.

Page 15, Line 393. Consider adding used as the after the Helium was

Page 15, Line 399. How many oil samples were analyzed?

Author Response

Response

Reviewer 2

We thank for our reviewer for the devoted work he/she has done and for the useful suggestions. The answers to the questions/comments are below.

Page 1, Line 17. It appears that an article is missing before the word aqueous. Consider adding an article (an).

Page 1, Line 17. Consider replacing "We formulated Pickering nanoemulsions" with "We formulated a Pickering nanoemulsion".

Page 1, Line 18. It appears that surface modified is missing a hyphen. Consider adding the hyphen (-).

Page 1, Line 18. It seems that agent may not agree in number with other words in this phrase. Consider adding an article (a) before the word stabilizing.

Page 1, Line 20. Consider adding the following words minimum inhibitory before MIC90 and put MIC90 inside parentheses.

Page 1, Line 22. It appears that Gram positive is missing a hyphen. Consider adding the hyphen (-).

Page 1, Line 22. It appears that Gram negative is missing a hyphen. Consider adding the hyphen (-).

Page 1, Line 23. It appears that the singular verb was does not agree with the plural compound subject metabolic activity and viability of Gram positive, Gram negative bacteria and Candida species, the generation of oxygen free radical species (ROS). Consider changing the verb to the plural form (were).

Page 1, Line 25. It appears that an article is missing before the word cellular. Consider adding an article (a).

Page 1, Line 28. The noun phrase donation seems to be missing a determiner before it. Consider adding an article (a).

Page 1, Line 36. Consider changing the noun the mixture to the plural form (mixtures).

Page 1, Line 43. It seems that preposition use may be incorrect here. Consider changing the preposition of with for.

Page 2, Line 53. It seems that there is a pronoun problem here. Consider adding the word itbefore impossible.

Page 2, Line 56. It appears that an article is missing before the word development. Consider adding an article (the).

Page 2, Line 59. Consider deleting the word techniques.

Page 2, Line 61. It seems that determiner use may be incorrect here. Consider adding the word the before case.

Page 2, Line 65. The spelling of solubilising is a non-American variant. For consistency, consider replacing it with the American English spelling (solubilizing).

Page 2, Line 73. It seems that determiner use may be incorrect here. Consider adding the word the before exact.

Page 2, Line 75. The spelling of stabilise is a non-American variant. For consistency, consider replacing it with the American English spelling (stabilize).

Page 2, Line 76. It appears that water in oil is missing two hyphens. Consider adding these hyphens (-).

Answer: All the suggested corrections were done in the revised text.

Page 2, Lines 78-80. Consider rewriting this sentence for a better understanding of its meaning (Previous studies have reported as a beneficial effect the decreased evaporation from o/w emulsions for nanoparticle-stabilized emulsions versus EOs-surfactant systems [21,22]).

Answer: We rephrased the text as follows:

Previous studies have reported decreased evaporation of EOs from o/w emulsion of nanoparticle-stabilized formulations versus EOs-surfactant systems to be a beneficial factor [21,22].

Page 2, Line 86. Consider deleting s from nanoemulsions and replacing one of the plausible modes of actions with a plausible mode of action.

Page 2, Line 89. Consider changing the title of subchapter 2.1 since is similar with the title with subchapter 4.1.

Page 2, Line 92. Consider adding (see 1. Figure) after TEM.

Page 3, Line 96. It seems that resolution may not agree in number with other words in this phrase. Consider changing the noun to the plural form (resolutions).

Page 3, Line 97. Consider changing the title of subchapter 2.2 since is similar with the title with subchapter 4.2.

Page 3, Line 98. Consider adding a before Pickering nanoemulsion.

Page 3, Line 100. The phrase In order to may be wordy. Consider changing In order to with To.

Page 3, Line 101. Consider deleting s from ones, adding an before emulsions, deleting s fromemulsions, and adding the before Tween80.

Page 3, Line 110. Consider deleting s from nanoemulsions.

Page 3, Line 111. Consider adding to the title of prepared emulsions after (MIC90).

Page 3, Line 113. The noun phrase growth seems to be missing a determiner before it. Consider adding an article (the).

Page 3, Line 119. Consider adding a before the word similar.

Page 5, Line 127. Consider changing the title of subchapter 2.4 since is similar with the title with subchapter 4.5.

Page 5, Line 128. Consider deleting the space between food and born (foodborne).

Page 5, Line 129. It appears that the singular verb has does not agree with the plural compound subject bacteria and fungi. Consider changing the verb to the plural form (have).

Page 5, Line 129. It appears that dose response is missing a hyphen. Consider adding the hyphen (-).

Page 5, Line 130. It seems that determiner use may be incorrect here. Consider adding a before two-fold.

Page 5, Line 132. It seems that figure may not agree in number with other words in this phrase. Consider changing the noun to the plural form (figures).

Page 6, Line 143. It seems that figure may not agree in number with other words in this phrase. Consider changing the noun to the plural form (figures).

Page 6, Line 145. It appears that the singular verb was does not agree with the plural compound subject the ROS (1085.86 ± 126.36), peroxide (1229.86 ± 164.52) and superoxide (1276.86 ± 165.42) generation. Consider changing the verb to the plural form (were).

Page 6, Line 145. The noun phrase effective increment seems to be missing a determiner before it. Consider adding an article (an).

Page 6, Line 149. It seems that determiner use may be incorrect here. Consider adding the word an before eight to nine-fold.

Page 6, Line 149. It seems that determiner use may be incorrect here. Consider adding the word a before two to four-fold.

Page 7, Line 154. Consider replacing 60 minutes with 1h.

Page 8, Line 161. It appears that time kill is missing a hyphen. Consider adding the hyphen (-).

Page 8, Line 163. It appears that an article is missing before the word case. Consider adding an article (the).

Page 8, Line 163. It seems that figure may not agree in number with other words in this phrase. Consider changing the noun to the plural form (figures).

Page 8, Line 165. It appears that an article is missing before the word case. Consider adding an article (the).

Page 8, Line 166. It seems that determiner use may be incorrect here. Consider adding a before two-fold.

Page 8, Line 166. It appears that an article is missing before the word killing. Consider adding an article (the).

Page 8, Line 169. It appears that colony forming is missing a hyphen. Consider adding the hyphen (-).

Page 9, Line 173. It appears that colony forming is missing a hyphen. Consider adding the hyphen (-).

Page 9, Line 177. It seems that figure may not agree in number with other words in this phrase. Consider changing the noun to the plural form (figures).

Page 11, Line 196. It appears that you are missing a comma after the introductory phrase After 1 hour of interaction. Consider adding a comma.

Page 11, Line 197. The noun phrase ethanolic solution seems to be missing a determiner before it. Consider adding an article (the).

Page 11, Line 199. The noun phrase amount seems to be missing a determiner before it. Consider adding an article (the).

Page 11, Line 202. The noun phrase ethanolic solution seems to be missing a determiner before it. Consider adding an article (the).

Page 11, Line 202. It appears that you are missing a point at the end of the sentence. Consider adding a point.

Page 11, Line 204. It appears that an article is missing before the word amount. Consider adding an article (the).

Page 11, Line 204. The noun phrase percentage seems to be missing a determiner before it. Consider adding an article (the).

Page 11, Line 207. It seems that determiner use may be incorrect here. Consider adding the word the before Effect.

Page 11, Line 214. It seems that the verb have does not agree with the subject. Consider changing the verb to the singular form (has).

Page 11, Line 215. It appears that an article is missing before the word effective. Consider adding an article (an).

Page 11, Line 218. The noun phrase conventional emulsion seems to be missing a determiner before it. Consider adding an article (the).

Page 11, Line 222. It seems that determiner use may be incorrect here. Consider adding the word the before intracellular.

Page 11, Line 228. The noun phrase higher local concentration gradient seems to be missing a determiner before it. Consider adding an article (the).

Page 11, Line 228. The noun phrase cell membrane seems to be missing a determiner before it. Consider adding an article (the).

Page 11, Line 231. The noun phrase easy escape seems to be missing a determiner before it. Consider adding an article (the).

Page 11, Line 232. The noun phrase conventional emulsion seems to be missing a determiner before it. Consider adding an article (the).

Page 12, Line 235. It appears that an article is missing before the word modified. Consider adding an article (the).

Page 12, Line 239. Consider replacing the word additional with another.

Page 12, Line 241. Consider replacing 28w/w% with 28% (w/w).

Answer: All the suggested corrections were done in the revised text.

Page 12, Line 243. Please verify that this propiltriethoxysilane is the correct name of the reagent used.

Answer: The correct chemical name was added.

Page 12, Line 245. The noun phrase Ethanolic solution seems to be missing a determiner before it. Consider adding an article (the).

Page 12, Line 245. The noun phrase modifying agent seems to be missing a determiner before it. Consider adding an article (the).

Page 12, Line 247. It appears that the singular verb was does not agree with the plural compound subject the ammonium hydroxide and ethanol. Consider changing the verb to the plural form (were).

Page 12, Line 247. The noun phrase reaction mixture seems to be missing a determiner before it. Consider adding an article (the).

Page 12, Line 248. Consider adding the acquisition country of the equipment used and a closing bracket after it.

Page 12, Line 256. Consider changing the title of subchapter 4.2 so as to include the preparation of conventional emulsion and ethanolic solution.

Page 12, Line 257. It appears that surface modified is missing a hyphen. Consider adding the hyphen (-).

Page 12, Line 261. It appears that an article is missing before the word emulsification. Consider adding an article (the).

Page 12, Line 272. Vancomycin (Van) was not mentioned in this section (4.3).

Answer: All the suggested corrections were done in the revised text.

Page 12. Nutrient agar media was not mentioned in this section (4.3).

Page 12. PBS was not mentioned in this section (4.3).

Answer: In section 4.3. a reference is given for our in-house nutrient agar media and also the source of PBS was given:

For fungi we used an in-house nutrient agar medium [36] while phosphate buffered saline (PBS, pH 7.4) was from Life Technologies Ltd., Hungary.

Page 12. You did not describe how you prepared CEt (mentioned in the abstract).

Answer: We included the following sentence into the text (section 4.2.)

To compare the different formulations ethanolic solution was also prepared; chamomile essential oil in 100 µg/mL concentration added to absolute ethanol, the solution was sonicated for 5 minutes.

Page 13, Line 278. Consider adding 90 after the word MIC.

Page 13, Line 287. The plural verb were does not appear to agree with the singular subject activity. Consider changing the verb to the singular form (was).

Page 13, Line 290. Consider adding the acquisition country of the equipment used.

Page 13, Line 291. Consider mentioning the test drugs used and their concentration range.

Page 13, Line 292. The plural verb were does not appear to agree with the singular subject activity. Consider changing the verb to the singular form (was).

Page 13, Line 294. It seems that ranges may not agree in number with other words in this phrase. Consider changing the noun to the singular form (range).

Page 13, Line 294. Consider mentioning the test drugs used and their concentration range.

Page 13, Line 303. Consider mentioning the test samples used. It seems that the verb were does not agree with the subject. Consider changing the verb to the singular form (was).

Page 13, Line 305. It appears that colony forming is missing a hyphen. Consider adding the hyphen (-).

Page 13. The number of replicates for each sample was not mentioned in this section (4.5).

Page 13, Line 314. Consider adding the model, the producer and the acquisition country of the equipment used (for centrifuge).

Answer: All the suggested corrections were done in the revised text.

Page 13, Line 317. Consider adding the acquisition country of the equipment used.

Answer: All equipment details were given in the revision.

Page 14, Line 321. It seems that determiner use may be incorrect here. Consider adding the word the before previously.

Page 14, Line 326. It appears that the phrase same volume does not contain the correct article usage. Consider adding the word the before same volume.

Page 14, Line 329. It seems that determiner use may be incorrect here. Consider adding the word the before distribution.

Page 14, Line 342. The plural verb were does not appear to agree with the singular subject Control. Consider changing the verb to the singular form (was).

Page 14, Line 343. It appears that colony forming is missing a hyphen. Consider adding the hyphen (-).

Page 14, Line 343. It seems that the verb were does not agree with the subject. Consider changing the verb to the singular form (was).

Page 14, Line 347. The noun phrase cell population seems to be missing a determiner before it. Consider adding an article (the).

Page 14, Line 352. Consider adding how many hundred microliters were used. It is one hundred?

Page 14, Line 353. It seems that the verb was does not agree with the subject. Consider changing the verb to the plural form (were).

Answer: All the suggested corrections were done in the revised text.

Page 14, Line 355. Consider adding the acquisition country of the equipment used.

Answer: All equipment details were given in the revision.

Page 14, Line 365. Consider replacing by one-way ANOVA test with using the One-Way ANOVA test.

Page 14, Line 365. It appears that P value is missing a hyphen. Consider adding the hyphen (-).

Page 15, Line 374. Consider adding the acquisition country.

Page 15, Line 376. The noun phrase mixture seems to be missing a determiner before it. Consider adding an article (a).

Answer: All the suggested corrections were done in the revised text.

Page 15, Line 378. The noun phrase magnetic stirrer seems to be missing a determiner before it. Consider adding an article (a).

Answer: The suggested correction was done in the revised text.

Page 15, Line 378. Consider adding the model, the producer and the acquisition country of the equipment used (for magnetic stirrer).

Answer: All equipment details were given in the revision.

Page 15, Line 378. It seems that determiner use may be incorrect here. Consider adding the word before rotational.

Answer: The suggested correction was done in the revised text.

Page 15, Line 378. Consider adding the model, the producer and the acquisition country of the equipment used (for rotational evaporator).

Answer: All equipment details were given in the revision.

Page 5, Line 379. It seems that preposition use may be incorrect here. Consider changing the preposition on with at.

Page 15, Line 393. Consider moving 0.25 µm before coating thickness.

Page 15, Line 393. Consider adding used as the after the Helium was

Answer: All the suggested corrections were done in the revised text.

Page 15, Line 399. How many oil samples were analyzed?

Answer: One oil sample was analyzed in duplicate.